# High-Throughput Sequencing of Diatom Community, Its Spatial and Temporal Variation and Interrelationships with Physicochemical Factors in Danjiangkou Reservoir, China

**Chunxia Zhang [1,2], Yuxiao He [1], Weiguo Li [1], Xiaoming Guo [1], Chunyan Xiao [1] and Tongqian Zhao [1,\*]**

1   Institute of Resources and Environment, Henan Polytechnic University, Jiaozuo 454000, China; zhangchxia@163.com (C.Z.); heyuxiao@hpu.edu.cn (Y.H.); wgli@hpu.edu.cn (W.L.); guoxiaoming@hpu.edu.cn (X.G.); xiaochunyan@hpu.edu.cn (C.X.)
2   School of Emergency Management, Henan Polytechnic University, Jiaozuo 454000, China
\*   Correspondence: zhaotq@hpu.edu.cn

**Abstract:** Diatoms constitute an important part of the phytoplankton community in lakes and reservoirs and play a significant role in regulating ecological balance. Danjiangkou Reservoir is the water source area of the middle route of China's South-to-North Water Diversion project. In order to explore the spatial and temporal distribution and know the governing factors of the diatom community, 18srRNA sequencing was carried out from seven sampling sites of the reservoir. At the same time, the concentration of nutrients present in the collected sample water was also determined. The results showed that a total of 51 genera and 96 species were thriving the community of diatoms in Danjiangkou Reservoir. *Discostella* was dominant in summer and autumn, accounting for 98.84% and 62.71% of the diatom abundance, respectively. *Aulacoseira* was dominant in spring and winter, accounting for 60.62% and 60.90%, respectively. *Discostella* and *Aulacoseira* showed significant differences in seasonal variation ($p < 0.05$). The colinear network of diatoms changed significantly with the seasons, mainly consisting of *Aulacoseira*, *Discostella*, and *Stephanodiscus*. RDA redundancy analysis showed that water temperature (WT), total nitrogen (TN), $NH_4^+$-N, pH, and electrical conductivity (Cond) were the main environmental factors driving the changes in diatom community structure.

**Keywords:** Danjiangkou Reservoir; high-throughput sequencing; temporal and spatial variation of diatoms; influencing factors

## 1. Introduction

In freshwater ecosystems, diatoms are usually the main primary producers in planktonic, periohytic, and epipelic communities of wetlands, and their abundance and composition can vary throughout the aquatic food web [1,2]. As major producers in the food chain [3], they fix atmospheric carbon dioxide through photosynthesis and constitute 20–25% of the primary productivity in water [4,5]. Diatoms generally reproduce quickly, have a short life cycle, and are sensitive to the aquatic environment (including physical and chemical changes in water) [6]. One big advantage of using them as biological monitors is that when cells die and fall to the bottom, the silica cell walls are left as a fossil record. They have the advantage of tracking the change in lake and reservoir properties dominated by environment and climate and thus become an important indicator for water quality detection [7–10].

The structure and composition of diatom communities vary with the intensity of environmental and anthropogenic factors [11,12]. Usually, water temperature is the main index of diatom community structure change [13–15], and different species of diatom show their responses to optimum temperatures for their best growth. Diatom growth is also affected by nutrients, rainfall, water depth and area of reservoirs, current velocity, wind force, and inter- and intraspecific community structure [16–20]. Among all the potential

factors, changes in water temperature and nutrients have the most direct impact on the diatom community. In China, the research on diatom community has focused mainly on lakes [21,22], and relatively little attention has been paid to artificial reservoirs with cross-climatic regions and water sources.

The methodology applied to enumerate the phytoplankton community structure actually governs the accuracy of the research results. So far, the qualitative identification of phytoplankton species and their cell densities present in a community has been accurately worked out by using traditional compound microscopy. However, this method needs a high level of professional skill from planktologists and at the same time could be regarded as time consuming. Therefore, the methods based on morphological observation usually underestimate the potential species diversity and community characteristics. However, high-throughput sequencing technology reduces the limitation of taxonomic expertise, especially for some small and fragile species of phytoplankton. With high-throughput sequencing of ribosomal DNA (rDNA) genes coupled with metabarcoding, the obtained data is more complete (based on 18RNA) and is widely used [23–25].

Danjiangkou Reservoir is the water source of the middle route of China's South-to-North Water Diversion project, which supplies water to cities along the northern route. Therefore, any change in water quality in the reservoir area will affect the health of residents and industrial development. Phytoplankton is a sensitive indicator of water quality detection, so the dynamic changes in the phytoplankton community in Danjiangkou Reservoir have received increasing attention. In some early studies on the phytoplankton community structure in Danjiangkou Reservoir, it was found that the phytoplankton diversity in the reservoir area increased significantly with the completion of the dam. The phytoplankton in the reservoir area are composed mainly of bacillariophyta and Chlorophyta, and the species of the former group dominate the population in spring, autumn, and winter. The main types of diatoms are *Cyclotella*, *Aulacoseira*, and *Fragilaria* [26,27]. The present research on phytoplankton in Danjiangkou Reservoir has focused mainly on studying qualitative and quantitative aspects based on microscopic observations. However, the seasonal distribution of phytoplankton in Danjiangkou Reservoir and its impact on the reservoir ecosystem are still not well understood, which may lead to a wrong understanding of the dominant species and community composition of phytoplankton.

In this paper, the diatom communities in the core water source area of Danjiangkou Reservoir were investigated in four seasons by using high-throughput DNA sequencing detection technology. The distribution of dominant diatom genera and species as groups was analyzed, and the environmental factors affecting diatom community structure in the reservoir area were identified. The aim of the study was therefore to reveal the seasonal variation and spatial distribution characteristics of diatom community structure in the reservoir area. This study will provide some reference for water quality monitoring and management of Danjiangkou Reservoir.

## 2. Materials and Methods

### 2.1. Study Area

This study was carried out in the Henan section of Danjiangkou Reservoir (111°30′22″–111°40′09″ E and 32°39′35″–32°52′24″ N), the middle route of the South-to-North Water Diversion project in Nanyang city, Henan Province, central China. The study area is located in the transition zone from the north subtropical climatic zone to the warm temperate zone. In this zone, the northeast wind prevails all year, and the annual average rainfall is 800–1000 mm, mainly intensive in summer. The average temperature is 26.22 °C and 3.68 °C in summer and winter, respectively, and about 16 °C in spring and autumn. The reservoir's annual water storage capacity reaches 29.05 billion $m^3$, and the normal water level is 170 m. The annual water volume is $130 \times 10^8$ $m^3$. Danjiangkou Reservoir covers an area of 1050 $km^2$, the study area accounted for 52% [28].

### 2.2. Sampling Point Setting and Collection of Samples

To collect representative samples from the Danjiangkou Reservoir, a total of 7 sampling points were set up covering the centre and peripheral area of the reservoir (Figure 1). The sampling points were Taocha (TC), Songgang (SG), Tumen (TM), Heijizui (HJZ), Kuxin (KX), Dangzikou (DZK), and Wulongquan (WLQ). The sampling was carried out in April, July, and October 2019 and January 2020 for the spring, summer, autumn, and winter seasons, respectively. Three parallel sampling points were set for each sampling point in the horizontal direction. The vertical direction of each point was a mixture of surface, 5 m, and 10 m water samples and filtered by 0.22 μm glass fiber filter (Shanghai Xingya). The filter membrane was shredded in a 10 mL centrifuge tube and stored in a liquid nitrogen tank at −80 °C.

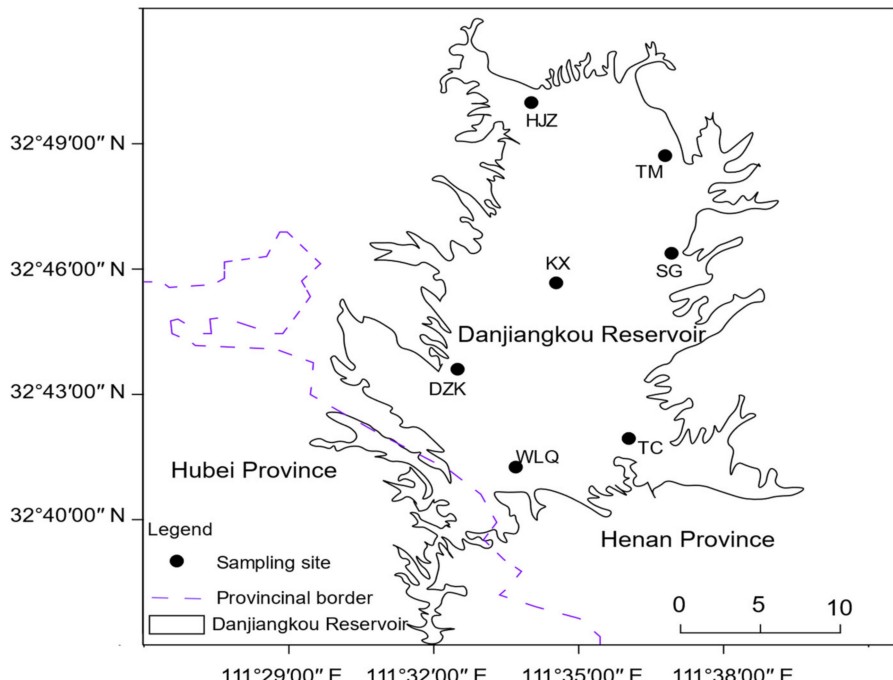

**Figure 1.** Setting of sampling points in the Henan section of the Danjiangkou Reservoir for physical and chemical index detection.

A YSI water quality analyzer (HQd Field Case) was used to measure the water temperature (WT, °C), pH, and electrical conductivity (Cond, ms/cm). The transparency of water (SD, m) was measured with the help of a standard Secchi disk. After pretreatment of the collected water samples, they were taken back to the laboratory for analysis of total nitrogen (TN, mg/L); ammonia- ($NH_4^+$-N, mg/L), nitrate- ($NO_3^-$-N, mg/L) and nitrite-nitrogen ($NO_2^-$-N, mg/L); and total phosphorus (TP, mg/L) and chlorophyll-a (Chl-a, mg/L). TN and TP concentrations were determined by standard ultraviolet spectrophotometry after digestion (UV-2600, Shanghai Dapu Instrument Co., Ltd., Shanghai, China); ammonia nitrogen and nitrate nitrogen were determined by the Nessler's reagent and ultraviolet spectrophotometry; nitrite nitrogen was determined by diazocoupled spectrophotometry; and the Chl-a was extracted by 90% acetone spectrophotometry. All parameters were recorded by the standard method [29].

### 2.3. DNA Extraction and Sequencing

Total DNA was extracted according to the instructions of an MP kit (FastDNATM SPIN Kit for soil), and the concentration and purity of DNA were detected by NanoDrop2000. Amplification PCR primers of diatoms were used to amplify the 18S rRNA V4 region. The primers used were DIV4F (GCGGTAATTCCAGCTCCAATAG) and DIV4R CTC TGATC-CAATGGATACAATA [30]. The PCR products were extracted by 2% agarose gel and

purified amplicons were pooled in equimolar amounts, and paired-end sequenced on an Illumina MiSeq PE300 platform (Illumina, San Diego, CA, USA) according to the standard protocols by Majorbio Bio-Pharm Technology Co., Ltd. (Shanghai, China).

### 2.4. Data Processing and Analysis

The 18S rRNA gene sequence data was analyzed by QIIME V 1.9.1, and sequences of length less than 50 bp after quality control was removed, so that the sequence could be clustered by the UPARSE software (version 7.1 http://drive5.com/uparse/ (accessed on 8 May 2022)) according to 97% similarity. The RDP classifier was used to classify and annotate each sequence, the comparison threshold was set to 70%, and the sequences unrelated to diatoms were removed [31,32].

The SPSS25.0 statistical software was used to carry out ANOVA based on the seasonal and spatial distribution of OTUs, and the heatmap clustering of diatom seasons was carried out using the software TBtools (Toolbox for Biologists) v1.09861. Redundancy analysis (RDA) is a constraint ranking method that directly links diatom assemblage with environmental variables; it was used to identify and evaluate impacts on the reservoir ecosystem. Pearson was used to analyse the correlation between genus and environmental factors. Mapping was performed with the R 4.1.0 software [33].

In order to investigate whether the distribution of diatoms in the four seasons followed the rule of seasonal succession, the symbiotic network of each season was constructed based on the relative abundance of OTUs. The OTUs with diatom abundance more than 0.1% were selected, and the Pearson's correlation coefficient of each pair of OTUs was calculated by the R 4.1.0 software. The threshold of interaction among species was determined, values of <0.6 in the correlation matrix were transformed into 0, and the seasonal nodes and edges were constructed using the molecular ecology software Gephi 0.9.2. The topological properties of seasonal collinear networks, including edges, nodes, degrees, clustering coefficients, short path lengths, modularity, and network diameters, were analyzed and compared.

### 2.5. Nucleotide Sequence Accession Numbers

In this study, all the Illumina MiSeq NCBI DNA sequence data were stored in a public reading archive (https://submit.ncbi.nlm.nih.gov/subs/sra (accessed on 8 May 2022)), biological project number: PRJNA820319, registration number: SUB11199378 (spring), SUB 11320234 (summer), SUB11251747 (autumn), SUB11252064 (winter).

## 3. Results

### 3.1. Physicochemical Properties of Danjiangkou Reservoir Subsection

The physicochemical parameters measured at all seven stations in four seasons of Danjiangkou Reservoir are presented in Table 1. The temperature fluctuated between 12.14 and 32.02 °C, and the average temperature in summer was 2.64 times that in winter. The concentration of $NH_4^+$-N started increasing from spring, reached its highest in summer, and gradually decreased in autumn and winter. TN concentration was lower in spring and autumn and slightly lower in winter than in summer. The seasonal distribution of nitrate nitrogen ($NO_3^-$-N) concentration was slightly different from that of $NH_4^+$-N and TN. $NO_3^-$-N concentration reached its lowest value in summer (0.47 mg/L) and its highest value in autumn (0.95 mg/L). Moreover, there was no significant variation in nitrate nitrogen concentration in spring and winter. On the whole, the physical and chemical properties of the water in the reservoir area showed seasonal changes in the four seasons, with higher pH, WT, TN, $NH_4^+$-N, and Cond in summer.

**Table 1.** Analysis of environmental factors in four seasons in Danjiangkou Reservoir.

| Environmental Factor | Spring | Summer | Autumn | Winter |
|---|---|---|---|---|
| pH | $8.80 \pm 0.05$ | $9.15 \pm 0.04$ | $8.76 \pm 0.05$ | $8.86 \pm 0.07$ |
| Cond/ms·m$^{-1}$ | $27.39 \pm 0.37$ | $28.21 \pm 0.69$ | $27.87 \pm 0.41$ | $27.39 \pm 0.40$ |
| WT/°C | $23.07 \pm 0.78$ | $32.02 \pm 0.90$ | $21.66 \pm 0.64$ | $12.14 \pm 0.57$ |
| SD/m | $5.37 \pm 1.03$ | $3.94 \pm 0.40$ | $3.32 \pm 0.32$ | $3.41 \pm 0.16$ |
| TP/mg·L$^{-1}$ | $0.03 \pm 0.00$ | $0.04 \pm 0.01$ | $0.04 \pm 0.01$ | $0.04 \pm 0.02$ |
| Chl-a/mg·L$^{-1}$ | $0.002 \pm 0.00$ | $0.005 \pm 0.00$ | $0.003 \pm 0.00$ | $0.002 \pm 0.00$ |
| NH$_4^+$-N /mg·L$^{-1}$ | $0.13 \pm 0.03$ | $0.17 \pm 0.03$ | $0.16 \pm 0.03$ | $0.13 \pm 0.02$ |
| NO$_3^-$-N/mg·L$^{-1}$ | $0.68 \pm 0.09$ | $0.47 \pm 0.07$ | $0.95 \pm 0.04$ | $0.68 \pm 0.10$ |
| TN/mg·L$^{-1}$ | $1.06 \pm 0.07$ | $1.26 \pm 0.24$ | $1.16 \pm 0.07$ | $1.19 \pm 0.10$ |
| NO$_2^-$-N/mg·L$^{-1}$ | $0.003 \pm 0.00$ | $0.003 \pm 0.00$ | $0.004 \pm 0.00$ | $0.002 \pm 0.00$ |

*3.2. Seasonal Variation of Diatom Community Structure*

A total of 231 OTU sequences belonging to 51 genera and 96 species were identified by Illumina MiSeq sequencing. There were some differences in the seasonal distribution of diatoms at the genus level (Figure 2A, the Supplementary Material Table S1). The distributional pattern of the number of diatom genera over the seasonal cycles followed 32, 29, 34, and 23 in spring, summer, autumn, and winter, respectively. The four seasons were detected by ANOVA based on OTU sequences, and the results showed obvious seasonal succession ($p < 0.001$). The diatom abundance in spring was much lower than that in other seasons, reached the highest in summer, and decreased gradually in autumn and winter. The dominant genera in the four seasons were mainly *Discostella* and *Aulacoseira*. Between them, *Discostella* was the dominant genus in summer and autumn and showed its absolute dominance in summer (98.84%). After a gradual decrease in autumn and winter (62.71% and 32.40%, respectively), the proportion of *Discostella* in spring was the lowest at only 1.99%. On the other hand, *Aulacoseira* was the dominant genus in spring and winter (60.62% and 60.90%, respectively), but it accounted for a low proportion in summer and autumn (0.35% and 14.52%, respectively). In addition, the proportion of *Cyclotella* in spring, at 21.28%, was second only to that of *Aulacoseira*, but *Cyclotella* was not dominant in other seasons. Based on one-way ANOVA, it was found that there were significant differences in the seasonal distributions of *Discostella*, *Aulacoseira*, and *Cyclotella*.

The seasonal variation of diatoms at the species level was not completely consistent with that at the genus level (Figure 2B). In the investigation, the number of diatom species occurring in spring, summer, and autumn were 53, 39, and 49, respectively, but only 29 species occurred in winter. The dominant species in the four seasons were *Discostella nipponica*, *Discostella woltereckii*, *Aulacoseira granulata*, and *Cyclotella ocellata*. Among them, *D. nipponica* was absolutely dominant in summer (96.60%) and decreased to 23.56% and 9.80% in autumn and winter, respectively. However, the lowest value recorded in spring was only 0.43%. *D. woltereckii* was the dominant species in autumn (39.14%) but decreased to 22.59% in winter and less than 1% in spring and summer. *A. granulata* was the first dominant species in spring and winter (60.62% and 60.84%, respectively), but had low proportions of 0.35% and 14.53% in summer and winter, respectively. *C. ocellata*, meanwhile, dominated only in spring at 20.57%. There were significant differences in the seasonal variation of the dominant species of diatoms ($p < 0.05$).

The seasonal variations in the diatom community were analyzed by Euclidean distance clustering analysis of the 30 OTUs with the largest abundance (Figure 3). The results showed that the four seasons' samples were obviously divided into four clusters. Among them, cluster 3 and cluster 4 included all samples in winter and autumn, while cluster 1 and cluster 2 included all samples in spring and summer. These results indicated that the diatom community structure had seasonal distribution characteristics.

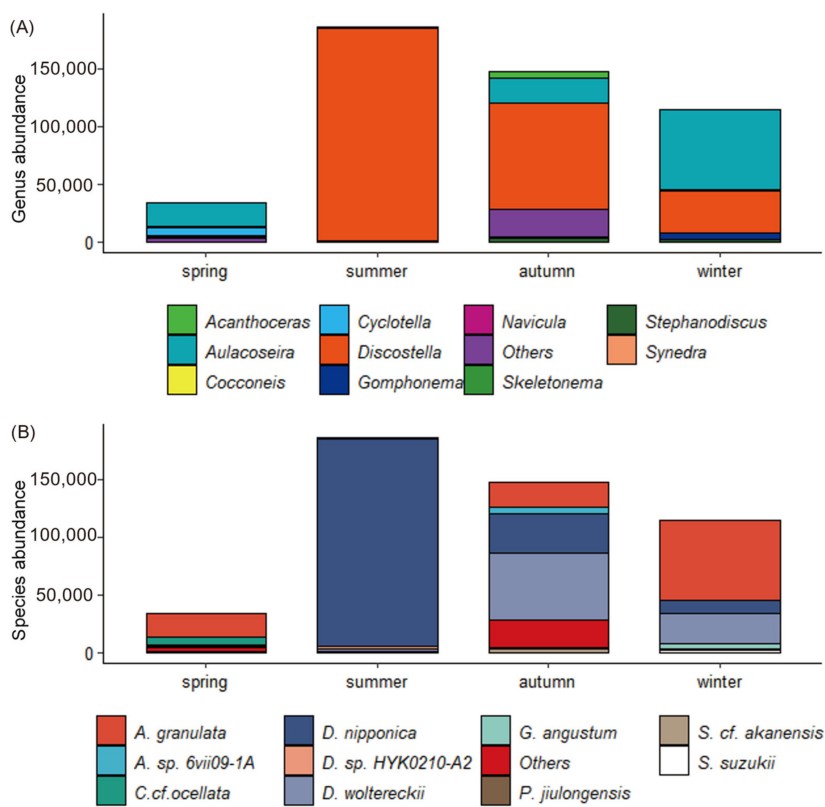

**Figure 2.** Seasonal distribution of diatom communities at genus and species levels in Danjiangkou Reservoir ((**A**) is genus and (**B**) is species).

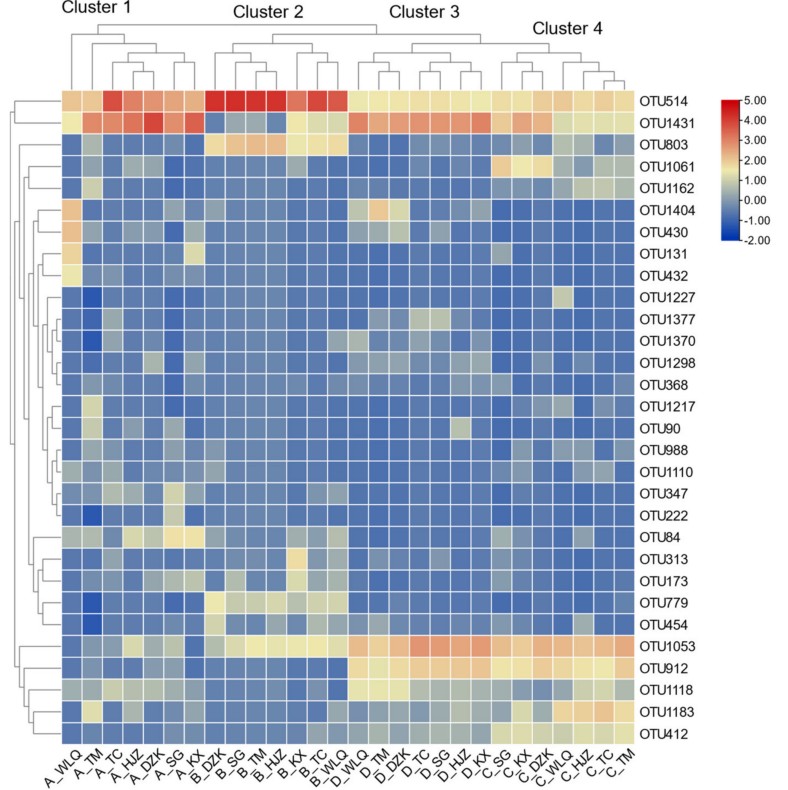

**Figure 3.** Four season cluster diagram of diatoms in Danjiangkou Reservoir (bottom horizontal axes: A is spring samples, B is summer samples, C is autumn samples, and D is winter samples).

### 3.3. Spatial Variation of Diatom Community

In order to test the spatial differences in diatom abundance at different sampling points, the sequences of OTU composition was analyzed by ANOVA. The results showed that there were significant differences in the diatom community among sites in spring ($p < 0.05$), but no significant variations in other seasons were found.

The diatom distribution between spring and winter was different to some extent. In spring (Figure 4A), the spatial distribution of *Aulacoseira* tended to decrease from northeast to west and south, while WLQ in the southwest was composed mainly of *Cocconeis*, *Cyclotella*, and *Gomphonema*, and the proportions of each were relatively uniform (about 23%). The distribution of *Aulacoseira* was the highest in the northeast TM of the reservoir area and lower in the southeast TC, where *Cyclotella* was mainly dominant. In winter, the abundance of *Aulacoseira* in the reservoir area was higher in TM (Figure 4D). *Discostella* dominated SG and TC in the east and southeast of the reservoir area, and *Aulacoseira* was dominant in the rest of the reservoir area.

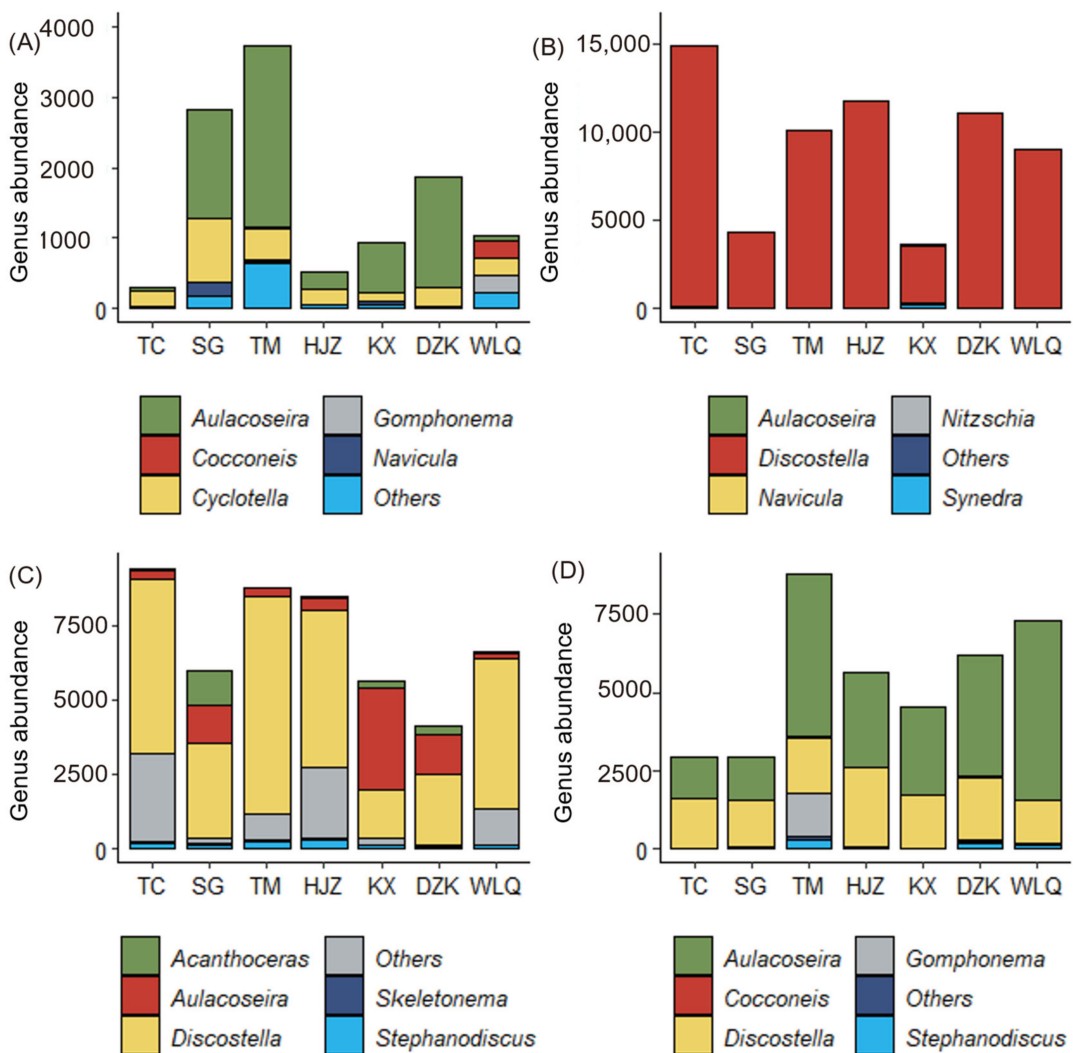

**Figure 4.** Spatial distribution of diatoms at the genus level in Danjiangkou Reservoir ((**A**) is spring, (**B**) is summer, (**C**) is autumn, and (**D**) is winter).

### 3.4. Network Relationship of Diatom Model Succession

In order to explore the succession of OTU abundance of diatoms in four seasons, a collinear network of four seasons was constructed (Figure 5). Different colors in the network represent different classes. Positive correlation accounted for more than 85% of

each season, while negative correlation existed only in autumn (14.29%). The numbers of network nodes and edges in spring were much larger than those in other seasons, and the average degree, average weighting degree, and graph density were the highest in summer, while the network diameter, modularization, average clustering coefficient, and average path length were highest in autumn.

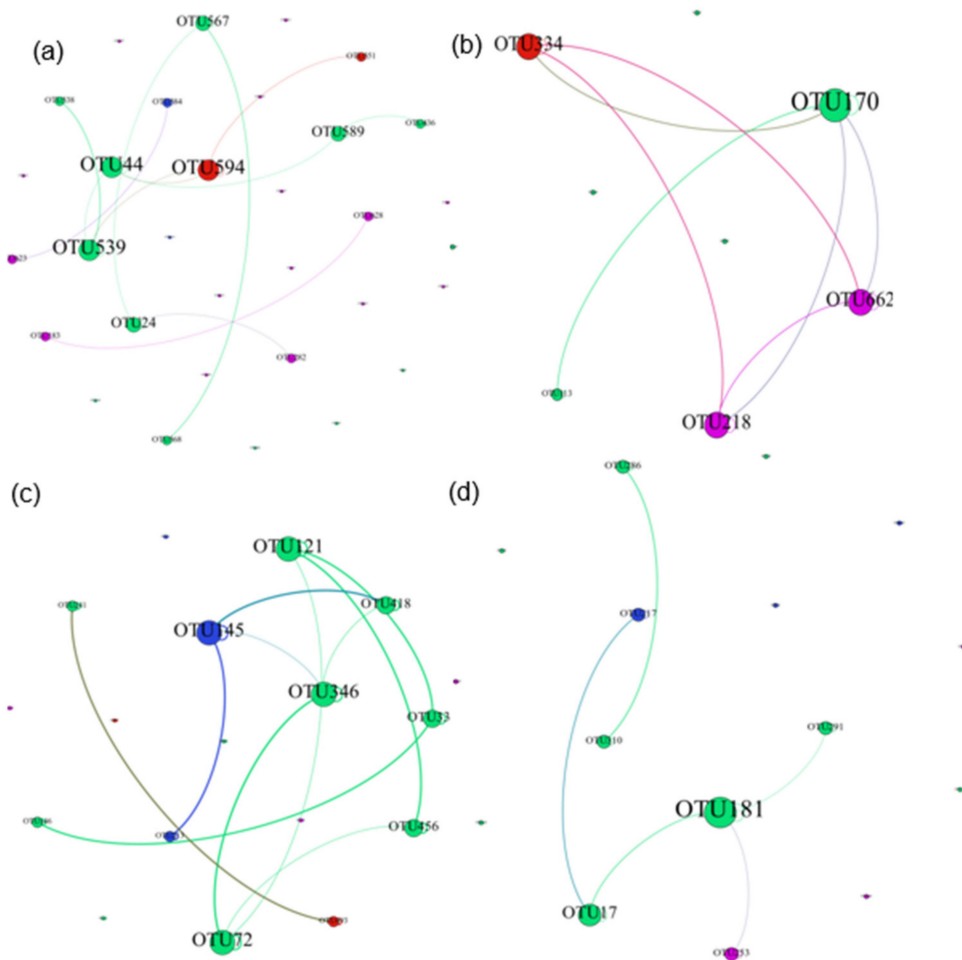

**Figure 5.** Seasonal collinear network of diatoms in Danjiangkou Reservoir ((**a**) is spring, (**b**) is summer, (**c**) is autumn, and (**d**) is winter).

In order to determine the network composition of each season, representative diatom species with high correlation in seasonal succession were correlated at the species level. In spring, *Stephanodiscus* cf *akanensis* (OTU539), *Asterionella formosa* (OTU594), *D. nipponica* (OTU589), and *A. granulata* (OTU44) showed a significant positive correlation. *A. granulata* (OTU170) had the largest number of connection edges. In summer, *Synedra ulna* (OTU334), *Navicula* sp. (OTU662), *Nitzschia palea* (OTU218), and *D. woltereckii* (OTU113) showed positive correlation. In autumn, the relationship was different from that in other seasons; there was a significant negative correlation between *D. woltereckii* (OTU121) and *A. granulata* (OTU 72), *Acanthoceras* sp. *6vii09-1A* (OTU346), and *unidentified diatom* (OTU145), and there were more connecting edges of *D. woltereckii* (OTU121) and *uncultured diatom* (OTU145). In winter, *Stephanodiscus suzukii* (OTU181) and *D. nipponica* (OTU17) had a significant positive correlation. In general, the colinear network of Danjiangkou Reservoir was a succession community based on *Aulacoseira*, *Discostella*, and *Stephanodiscus*.

### 3.5. Environmental Factors Affecting Diatom Seasonal Variation

To explore the relationship between the diatom community and environmental factors, the top 10 dominant genera and environmental factors in the four seasons were selected for CCA or RDA analysis. According to the test, the gradient length of the four-season sample to the response data was less than 3, so RDA was selected for redundancy analysis (Figure 6). After adjustment, the explanatory degree of variance was 65.3%, and the accumulative characteristic values of the first two axes could explain 76.43% of the species variation.

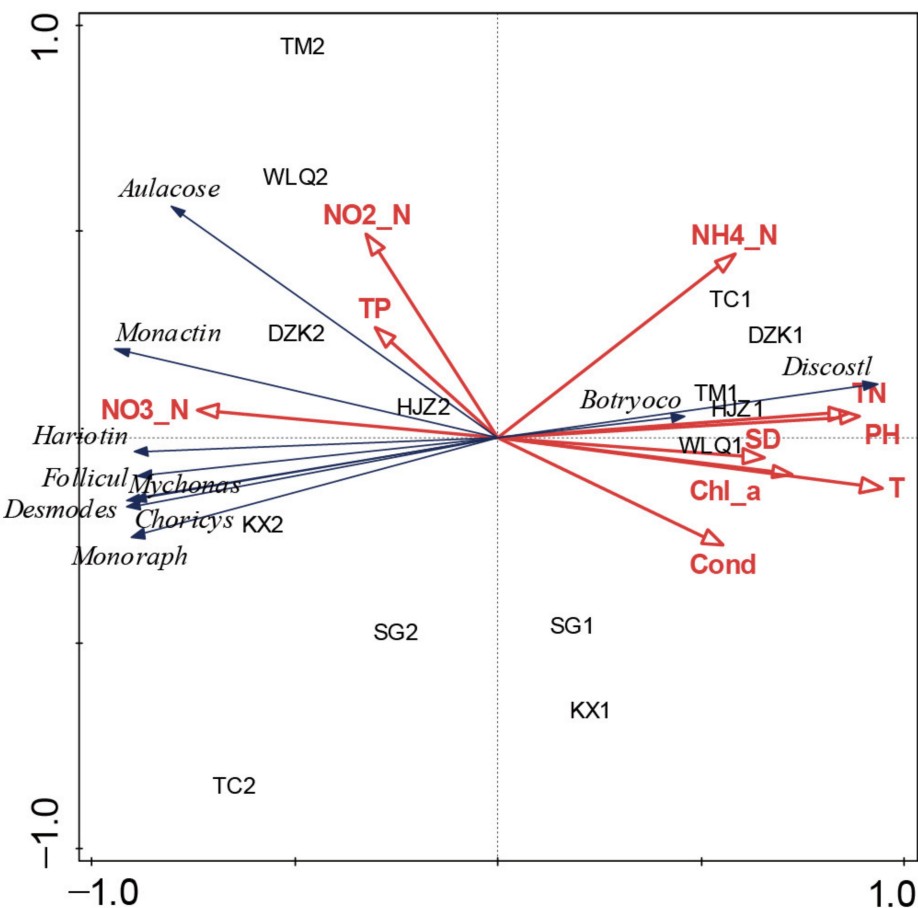

**Figure 6.** RDA diagram of dominant diatom genera and environmental factors in the Danjiangkou Reservoir (1 is spring, 2 is summer, 3 is autumn, and 4 is winter).

On the RDA diagram, the species had obvious seasonal distribution characteristics. Different sampling sites gathered together according to the season. On the top left of Figure 6 are spring sampling sites, and on the lower left are winter sampling sites. *Aulacoseira* was a representative species in two seasons. On the right side of the picture are mainly summer sampling sites, and the representative species was *Discostella*. On the other hand, the autumn sampling points gathered in the central part of the RDA map span four quadrants, representing the dominant genera of *Discostella* and *Aulacoseira*.

In addition, RDA analysis showed that WT, TN, $NH_4^+$-N, Cond, and pH had significant effects on diatom community structure in four seasons ($p < 0.05$), explaining a total of 72.2% of species variation. Correlation analysis of species and key environmental factors was conducted per Pearson. The results showed that five environmental factors were positively correlated with *Discostella* ($p < 0.01$). *Aulacoseira* was negatively correlated with WT and Cond ($p < 0.001$ and $p < 0.05$), and *Stephanodiscus* showed negative correlation with pH and WT.

## 4. Discussion

A relevant literature study showed that Danjiangkou Reservoir is in a state of medium nutrition concentration [34,35]. Bacillariophyta and Chlorophyta are the dominant phytoplankton algae in the reservoir. Diatoms play a vital role in the health of the aquatic ecosystem, and changes in the physical and chemical parameters in the reservoir ecosystem can significantly change the community composition of diatoms [32]. Some species have specific nutritional requirements under a wide range of nutrient conditions of water. These have become an indicator to judge the nutrient concentration, nutrient supply rate, and water quality of water bodies [36,37]. High-throughput sequencing can detect microscopic algae that are not easily identified by ordinary microscope observation. These algae may dominate the water body and play an important role in the phytoplankton community structure in the water body.

### 4.1. Temporal and Spatial Variation of Diatom Community Structure in Danjiangkou Reservoir

In this study, the seasonal distribution of the dominant diatom genera, namely *Discostella* and *Aulacoseira*, was related to the change in water temperature in the reservoir area. Between them, *Discostella* was the dominant genus in summer and autumn, while *Aulacoseira* was more dominant in spring and especially in winter. The relative abundance of *Discostella* was the highest in summer, reaching 98.84%. As the temperature decreased, the proportion of *Discostella* in winter decreased sharply to 14.53%, indicating that *Discostella* is more suitable for growing in a higher temperature environment. *Aulacoseira* accounted for only 0.35% in summer, but in winter, the proportion of *Aulacoseira* increased to 60.84%, indicating that *Aulacoseira* is more suitable for growth at lower temperatures. This was consistent with results from Lake Tahoe in the United States and Lake Cheongpyeong in South Korea [38,39].

In addition to temperature, the dominant genus of diatoms and their spatial differences may also be affected by the East Asian monsoon. In this study, there was little difference in reservoir temperature between spring and autumn (less than 2 °C). The dominant genus is *Aulacoseira* in spring and *Discostella* in autumn. Danjiangkou Reservoir is dominated by the northeast wind all year. A study of the diatom community deposited in Huguang Maar Lake in south-eastern China showed that the East Asian monsoon system controlled the prevailing climatic conditions and that the seasonal difference of the dominant genus of diatoms was closely related to the limnology in this area [40]. The lake area was dominated by *Aulacoseira* in the strong monsoon period and *Discostella* in the weak monsoon period. A study of seasonal sediment traps in the same lake where this change was recorded further supported the trend in the classification of these dominant genera [41]. During the study period, the wind strength in spring (3.92 m/s) was significantly higher than that in autumn (2.80 m/s) (wind speed data from the European centre for medium-range weather forecasts (ECMWF) (https://cds.climate.copernicus.eu/ (accessed on 1 January 2020))), which probably affected the distribution of *Aulacoseira* and *Discostella* in Danjiangkou Reservoir.

In terms of spatial distribution, *Aulacoseira* had higher distribution in TM in the northeast of the reservoir area in spring and winter, while in the southeast of the reservoir area, *Cyclotella* in spring and *Discostella* in winter were dominant. The variation in the spatial distribution of phytoplankton in the reservoir area was also affected by the perennial northeast wind. In Danjiangkou Reservoir, *Aulacoseira* was more distributed in TM in the northeast of the reservoir area under the action of wind in spring and winter, while TC in the southeast of the reservoir was less affected by wind. *Cyclotella* and *Discostella* were more likely to accumulate in TC in the southeast of the reservoir because of their small particles floating with the water flow. In summer and autumn, as the wind weakened, the changes in the spatial distribution were no longer significant.

Collinear network analysis showed that besides *Discostella* and *Aulacoseira* as dominant genera, *Stephanodiscus* had about 45.3% abundance in early spring in Danjiangkou Reservoir [42]. *Stephanodiscus* often forms a dominant member of the phytoplankton pop-

ulation in late winter and early spring, consistently with a study of diatoms in Yeongsan River in South Korea in winter and a study of sedimentary diatoms in Baikal Laken [43,44]. Collinear network analysis also showed that there were more negative correlations among autumn species than in other seasons, which reflects that the relationships between species was more complex.

### 4.2. Factors Affecting the Diatom Community in Danjiangkou Reservoir

The factors affecting diatom community change include nutritional concentration, wind power, water temperature, light, reservoir size and depth, composition of food web and interaction among species, all of which can affect the composition of diatom groups. Among these, water temperature was the main factor affecting the diatom community in Danjiangkou Reservoir. Water temperature can indirectly affect the composition of diatom community structure by regulating chemical and biochemical processes, zooplankton, bacteria, and fish [45]. In addition, nutrient input is an important controlling factor, because nutrients can be directly absorbed by diatoms and thereby affect the diatom population [46].

RDA analysis showed that WT, TN, $NH_4^+$-N, Cond, and pH were the main factors driving the succession of diatoms in Danjiangkou Reservoir. An RDA diagram and Pearson correlation analysis showed that *Discostella* was significantly and positively correlated with those five environmental factors ($p < 0.01$). The abundance of *Discostella* was highest in summer (98.84%), and the main environmental factors, WT, TN, $NH_4^+$-N, Cond, and pH, were highest in summer, indicating that the environmental conditions in summer were more conducive to the relative abundance of *Discostella*. $NH_4^+$-N and TN were especially important. $NH_4^+$-N can be directly absorbed by algae, so it is the preferred nitrogen for algae to absorb [47,48]. It was reported that *Discostella* cell density was high in the late stage of freezing and reached a peak in August in the oligotrophic Jordan Pond in the United States. In addition, *Discostella* was more abundant in the epilimnion under nitrogen-limited conditions [49,50].

In the present investigation, the population density of *Aulacoseira* showed a significant negative correlation with WT ($p < 0.05$), which further verified its adaptation to low-temperature water environments for achieving optimum growth. This was consistent with studies of Lake Poyang in China and Lake Tahoe in the United States [39,51]. By analysis of the diatom sediment cores in Lake Tahoe, United States, it was found that in the little ice age region, diatoms were characterized by *Aulacoseira*, while in the transition zone, *Discostella* abundance increased rapidly with climate warming and anthropogenic influences. Pearson correlation analysis showed that *Aulacoseira* and *Discostella* were significantly negatively correlated ($p < 0.05$). In addition to the above factors, there was a significant negative correlation between *Aulacoseira* and Cond, showing that too-high Cond was disadvantageous to the growth of *Aulacoseira*. This is a problem that should be paid attention to in related experiments in the future.

*Aulacoseira* and *Discostella* were dominant in Danjiangkou Reservoir, and this was related to the nutrient concentration of water body. It was found in a previous study that diatom composition changes in lake ecosystems are usually due to the process of artificial eutrophication [37]. During 1959–2012, the N/P ratio of Dashahe Reservoir continually increased over time, and the increasing N/P ratio appeared to stimulate nitrogen-tolerant/-loving diatoms while inhibiting phosphorus-limited/-essential diatoms. In particular, *Aulacoseira* and *Discostella* were dominant in nitrogen-rich environments. The distribution of diatoms was the result of changes in the nutrient resource supply patterns in Dashahe Reservoir in recent decades [37,52]. Danjiangkou Reservoir is similar to Dashahe, and the total nitrogen concentration of Danjiangkou Reservoir has been relatively high for a long time [34,35]. Meanwhile, the phosphorus concentration is relatively low [26,30,31]. A higher N/P ratio is more conducive to the growth of nitrogen-tolerant/-loving phytoplankton, but the specific generation background needs to be further studied in combination with the reservoir environment.

The climate of Danjiangkou Reservoir is located in the transition zone from subtropical to warm temperate, and the diatom community structure is different from that of single climatic zone. The climate in summer is similar to that in the tropics, but the temperature in winter is significantly lower than that in the tropics. Therefore, the seasonal distribution of diatoms in Danjiangkou Reservoir has the common characteristics of two seasons. Because of the elevated water temperature in summer, diatoms possess the characteristics of tropical algal communities. The dominant diatom was the small-celled *Discostella* (98.84%), of which the dominant species, *D. nipponica* (96.60%), is 3–4 μm in cell diameter [53]. *D. nipponica* was overwhelmingly dominant in summer, which had obvious seasonal differences with winter. In winter, the diatom community was mainly composed of *A. granulata* (cell diameter 4.5–21 μm, height 5–24 μm). The results of this study were similar to those drawn from a trait-based model of temperature and tropics regions of the Atlantic Ocean phytoplankton communities [54]. Based on a data-driven algae characteristic model, differences in the characteristics of phytoplankton communities were found between the mid-Atlantic temperate zone and the tropical zone. The results showed that the stability of the tropical environment inhibited the potential seasonal fluctuation of the average cell size of phytoplankton, resulting in a continuous decrease in cell size (<1.3 Logmm ESD). In contrast, the seasonally pulsing environment in the temperate region enhanced nutrient uptake and utilization, pushing mean cell size and size variance to higher values (about 3.2 Logmm ESD) [54,55].

### 4.3. The Effect of Detection Technology on Research Results on Diatoms in Aquatic Ecosystems

In the past decade, high-throughput sequencing technology has been widely applied in the study of the structure of the planktonic community and has been proven to be efficient for biodiversity monitoring and quantification in aquatic ecosystems. In this study, we used high-throughput sequencing technology to amplify 18S rDNA V4 region DNA to investigate the diatom community structure in Danjiangkou Reservoir. To the best of our knowledge, this was the first report on the diatom community patterns in this important water source.

The detection of phytoplankton composition by traditional morphological methods provided valuable data for us to understand the changes in the phytoplankton community structure in Danjiangkou Reservoir. There are some differences between traditional morphological methods and high-throughput sequencing techniques in the detection of diatoms in Danjiangkou Reservoir. In the investigation before the construction of Danjiangkou Reservoir in 1958, diatoms such as *Cyclotella*, *Synedra*, and *Nitzschia* were dominant in the southern rivers, while dinoflagellates, Chlorophyta, cyanobacteria, and diatoms (mainly *Nitzschia*) were dominant in the Danjiang River [56]. After the reservoir was built in 1968, it was found that the species composition of phytoplankton in Danjiangkou Reservoir gradually evolved from river-type diatoms to the diatom–dinoflagellate–cyanobacteria type, and the dominant genera of diatoms were mainly composed of *Aulacoseira*, *Synedra*, *Navicula*, *Fragilaria*, and *Cyclotella* in 1992 and 1993 [57]. In 2017–2019, the phytoplankton in the reservoir area were mainly diatoms, Chlorophyta, and cyanobacteria, and diatoms were mainly composed of *Cyclotella*, *Aulacoseira*, *Achnanthes*, *Nitzschia*, and *Synedra* [28,35]. In this study, a total of 51 genera and 96 species were identified by Illumina MiSeq sequencing. The number of species identified in this investigation was higher than that of the 19 genera and 63 species of diatoms observed under microscope in Danjiangkou Reservoir [26]. Therefore, the research results greatly supplemented the species information in the phytoplankton community. In comparison with morphological methods, DNA metabarcoding is a sophisticated molecular tool that is linked to morphologically identified species because of its relatively high sensitivity and stability [25,58]. For example, the morphological traits of some species are easily affected by the environment, probably leading to biases in classification by microscope observation. Moreover, some species with extremely low abundance are not easily found by traditional methods [30,59,60]. In this study, *Discostella* was the dominant genus in summer and autumn in Danjiangkou Reservoir, with species such as



*D. nipponica* (diameter 3–4 μm) and *D. woltereckii* (diameter 6.5–7.5 μm) [53,61], which have not been reported before. One possible reason is that their extremely low abundance was easily neglected by microscope observation in previous studies. In the future, the impact of *Discostella* on the ecology of the reservoir area needs to be further studied.

## 5. Conclusions

In this study, the diatom community in Danjiangkou Reservoir was analyzed in terms of temporal and spatial variation for the first time. Throughout the course of a one-year study, the dominant genera of diatoms in the reservoir were mainly *Aulacoseira* in spring and winter and *Discostella* in summer and autumn. Through the analysis of RDA redundancy, it was revealed that WT, TN, $NH_4^+$-N, Cond, and pH were the main environmental factor driving change in the structure of the diatom community. The diatom *Discostella* detected in this study was a dominant genus that had not been found in the Danjiangkou Reservoir before. *Discostella* and *Aulacoseira* were dominant in the reservoir area; this was related to the excess of nitrogen concentration in the reservoir area. Therefore, controlling nitrogen input and reducing the N/P ratio would be effective management methods to restore the ecosystem diversity and improve the water quality of Danjiangkou Reservoir.

**Supplementary Materials:** The following supporting information can be downloaded at: https://www.mdpi.com/article/10.3390/w14101609/s1, Table S1: Seasonal distribution of genus and species of diatoms in Danjiangkou Reservoir.

**Author Contributions:** Conceptualization, X.G., W.L. and T.Z.; methodology, W.L., C.Z. and T.Z.; software, Y.H. and C.Z.; validation, Y.H., W.L. and T.Z.; formal analysis, Y.H., C.X. and C.Z.; investigation, C.Z.; resources, T.Z.; data curation, W.L. and T.Z.; writing—original draft preparation, C.Z.; writing—review and editing, C.Z., C.X., W.L. and T.Z.; visualization, X.G. and C.Z.; supervision, W.L. and T.Z.; project administration, W.L. and T.Z.; funding acquisition, T.Z. All authors have read and agreed to the published version of the manuscript.

**Funding:** This research was funded by the National Natural Science Foundation of China, grant number U1704241. The Plan for Scientific Innovation Talent of Henan Province, grant number 194200510010.

**Institutional Review Board Statement:** Not applicable.

**Informed Consent Statement:** Not applicable.

**Data Availability Statement:** The datasets generated and analyzed during the current study are available from the corresponding author on reasonable request.

**Acknowledgments:** The authors are very grateful to Yuxiao He and Weiguo Li's graduate students for accompanying me to take samples in the field. I also thank Hongqi Meng for his help in the experiment.

**Conflicts of Interest:** The authors declare no conflict of interest.

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
