# Peer review of "High-Throughput Sequencing of Diatom Community, Its Spatial and Temporal Variation and Interrelationships with Physicochemical Factors in Danjiangkou Reservoir, China"

_water, doi:10.3390/w14101609_

Round 1
Reviewer 1 Report
While I should start by saying that this manuscript is one of the finest examples I've seen of using metabarcoding to answer actual ecological questions rather than simply constructing a genetic flora, there are still some issues that must be resolved before publication. There are a number of issues with grammar and clarity throughout the manuscript (see specific comments below), but these can likely be resolved easily. There are also some inferences made (particularly in the Discussion) that are poorly-supported and must be addressed, also outlined below. Once these are resolved, I look forward to seeing this manuscript in print.
Abstract
Lines 11-12: “Diatoms constitute an important part of the phytoplankton community in lakes and reservoirs and play a significant role in regulating ecological balance.”
Line 14: “…know the governing factors of the diatom community…”
Line 16: “…same time, the concentration of nutrients…”
Line 17: “The results show that a total of 15 genera…”
Line 18: “…accounting for 98.84% and…”
Introduction
Line 60: “metabolic coding” This is a vague term. Please explain in specific terminology what you mean by this.
Lines 65-67: “…the dynamic changes of the phytoplankton community…have been widely concerned.” “Concerned” is probably a poor word choice here. I get the idea from context, but this sentence is awkwardly phrased.
Line 73: “…Reservoir has been focused mainly on the qualitative and quantitative aspects…”
Line 75: Consistency in capitalization: Danjiangkou Reservoir
Line 75: How many of the “dominant” diatom species listed above fall into this category of “<5 um”? There are plenty of Cyclotella, Aulacoseira and Fragilaria species which are larger than 5 um.
Line 79: “high throughput” what? High throughput DNA sequencing?
Line 82: “seasonal evolution” suggests a permanent change. “Seasonal variation” is a better term.
Line 83: “It provides…” What provides reference: this study or the diatom community structure?
Methods and Materials
Line 93: “The average temperature is 26.22oC and 3.68oC in summer…”
Line 122: “All parameter by standard method…” It seems like there is a missing verb…all parameters recorded by standard method, perhaps?
Lines-124-125: “The mean for each of the physicochemical parameters measured at all seven stations of Danjiangkou Reservoir has been presented in Table 1 for four seasons of the year of the study.”
Line 132: Where did the MiSeq sequencing occur (institution)?
Line 137: “RDP classifier was utilized to annotate each sequence, using a comparison threshold of 70%, and the sequences unrelated to algae were removed.” Any algal group? Or just diatoms? What was the identity threshold to identify OTUs to taxa?
Results
Line 162: “…fluctuated between 12.14oC and 32.02oC…”
Line 169: “…changed a little in spring and winter.” Changed how? Increase? Decrease? Changed relative to summer and autumn or changed between spring and winter?
Line 175: “Seasonal evolution…” Again, this suggests a permanent change. “Seasonal variation” is probably a better term.
Line 176: please reference the Supplementary Material for species identification
Remove underscores between genus and species
Figure 2: Some of the colors are difficult to distinguish—particularly Aulacoseira and Cyclotella in 2a. Figure caption: Seasonal and spatial distribution of diatom communities at genus (a) and species (b) levels in Danjiangkou Reservoir
Lines 210-213: “Euclidean” what? This is an awkward paragraph of sentence fragments and missing capitalization. “Obvious” might be the wrong term for a heat map which is overwhelmingly one color. Perhaps the authors could indicate some specific OTUs which vary by cluster to illustrate this variation.
Figure 3 caption: there are two horizontal axes on this figure, at the top and bottom
Lines 219-222: This is also a bit awkwardly phrased, as the stated ANOVA was to test for differences between sites, but the results are presented by seasonality.
Figure 4: Again, there is some difficulty distinguishing colors, particularly Aualcoseira v. Navicula in (a), Acanthoceros v Skeletonema in (c) and Aulacoseira v others in (b) & (d)
Line 232: “…with the same distribution characteristics as in spring (Figure 4(d))” Really? Because the abundance of Aulacoseira in sites TC, HJZ and WLQ all appear measurably higher and measurably lower in site SG between Fig. 4D than Fig. 4A
Lines 238-239: What is the source of the classification scheme outlined here, because this sentence does not correspond to the results shown in Figure 4, where Aulacoseira (Coscinodiscophyceae) and Cyclotella (Mediophyceae) appear dominant in the spring and the summer is dominated by Discostella (Mediophyceae). The Bacillariophyceae only appear in significant abundance at site WLQ in the spring.
Line 271-272: “…obvious seasonal distribution characteristics in space.” Which space?
Line 274-275: “Besides Aulacoseira, the representative species in spring was also observed via the distribution of Cyclotella, Navicula, and Cocconeis.” Another awkwardly phrased sentence. Do the authors mean that the presence of Cyclotella, Navicula and Cocconeis were indicative of a spring flora? Or is somehow the distribution of those genera across the RDA space indicate another species as representative of spring?
Figure 6 and caption: The caption does not do a very good job describing the figure. Seasonality is expressed as capital letters according to the caption, but in the figure it looks like seasonality might be represented by the shapes of the plot points. Which is it?
Discussion
Line 291: “…a state of medium nutrition.” Nutrition or nutrient concentration?
Line 292: These groups might represent the dominant phytoplankton—benthic algae were not explored.
Line 293: “Because diatoms play a vital role in the health of the aquatic ecosystem.” Sentence fragment.
Line 295: Environmental factors are influencing the growth types of diatoms (planktonic, benthic, coloniality) or the assemblage of diatoms (taxa present)? This study suggests the latter.
Line 299-300: Microscopic algae that are not easily observed in an “ordinary microscope” dominate the phytoplankton community structure? The authors provide us no indication of the size or resolvability of the dominant taxa in this study. Aulacoseira, in fact, is quite recognizable in the compound microscope.
Line 301: Again, “evolution” suggests a permanent change
Line 305: “growth proportion”? I suspect the authors mean “relative abundance” of the taxa in regard to the rest of the assemblage, not anything related to growth of the cells.
Lines 322-324: Wind strength was not among the tested environmental factors. Would increased rainfall associated with monsoon season affect any factors that were tested, such as pH or conductivity?
Line 336: What specific “certain proportion” does Stephanodiscus occupy?
Line 341: Symbiosis seems like a bit of a leap. Other than a correlation in relative abundance, is there any evidence to suggest an association as strong as symbiosis?
Line 343: Again, “nutritional status” or “nutritional concentration”? The latter is what was measured in this study
Line 357: Again, what was measured was relative abundance of the taxa, not growth rate
Lines 358-362: I’m not sure what the authors are trying to say in this passage. Was the relative abundance of Discostella in Jordan Pond highest at the beginning of summer (“…late stage of freezing…”?) or the end (“…a peak in August.”). Was the relative abundance of Discostella highest in the surface layer in August but higher in other layers of the water column in July?
Line 363: “…growth environment…” Again, growth wasn’t measured—relative abundance was. Growth rates might increase for all taxa in the summer, but the relative abundance of Discostella and Aulacoseira were altered in relation to the rest of the assemblage, correct?
Line 367: “By analyzed…” Sentence fragment, and should begin “By analysis of…”
Line 375: “Aulacoseira and Discostella are dominant in Danjiangkou Reservoir and this is related to the nutrient concentration of the water body.”
Lines 376-378: “In the study of phytoplankton in Dashahe Reservoir [insert citation], it was found that the change of lake ecosystem is usually attributed to the process of man-made eutrophication.” What specific change?
Line 381: “…Coscinodiscophyceae of diatoms, Aulacoseira and Discostella, are…” Again, Discostella is in the Mediophyceae
Line 382-383: citation needed
Line 383-385: “Danjiangkou Reservoir is similar to Dashahe in that the total nitrogen in Danjiangkou Reservoir has exceeded the level 3 standard of surface water environmental quality, while the phosphorus concentration is relatively low [26, 29, 30].” What is “level 3 standard”?
Line 389: “The climate of Danjiangkou Reservoir is in the transition zone from subtropical to warm…” I’m assuming Danjiangkou Reservoir is not the only lake or reservoir in this zone.
Lines 394-398: What are the “characteristics of a tropical algal community”, because the only citation that follows looks at temperate and tropical communities
Lines 398-399: Two sentence fragments
Lines 396-403: “small-celled” vs “large-celled” These are all relative terms without any context. What is the size range of “small” vs “large”? Species with a diatom genus can vary widely in cell size as well, so how large are the taxa detected in this study?
Line 409: If this is the first report on diatom community patterns, what did the 1958 study cited in the next paragraph study?
Lines 411-430: I have a couple of issues with this passage. For one, the genus Discostella was described from transferred Cyclotella species in 2004 (Houk & Klee 2004, Diatom Research vol. 19, issue 2), so the 1958 study could not have resolved Discostella from the assemblage. The 1958 study, however, did apparently suggest that Cyclotella dominated the assemblage. Were the Cyclotella species documented in 1958 the same species that would later be transferred to Discostella? This might not be a different result than found in the current study. Additionally, several of the taxa identified in this study and listed in the Supplementary materials are taxa associated with marine environments (Pseudo-nitzschia, Skeletonema, Fragilariopsis, Licmophora, Actinocyclus)—were these actual “detected species” or the result of some sort of artifact of the eDNA and metabarcoding techniques? Is “species detection” really “improved” if the incorrect species are identified?
Author Response
We are very grateful that you kindly give us a major revision for our manuscript (water-1665698) to meet with the published standard in journal water. Thanks for the reviewers’ comments concerning our manuscript. Those comments are all valuable and very helpful for improving our paper. We have studied comments carefully and have made major corrections according to reviewers’ suggestions. A point-by-point response letter has been provided. All revised portions are indicated in the text by highlighting in the revised manuscript. We re-edited the manuscript, and language of revised manuscript has been improved. Please see the attachment.

Reviewer 2 Report
Review: High Throughput Sequencing of Diatom Community, its Spatial and Temporal Variation and Interrelationships with Physicochemical Factors in Danjiangkou Reservoir, China
The article is very well written. It contains valuable new information on the diatom composition of the Danjiangkou Reservoir. The applied eDNA technic s a valid and emerging tool as well as the analytical techniques and statistics used throughout the manuscript. I didn't find scientific mistakes and problems that corrections would improve the manuscript.
Author Response
Thank you for your recognition of our work and evaluation of the paper.
Round 2
Reviewer 1 Report
I thank the authors for their improvements to the manuscript, which makes for an easier read. There are only a few points from my previous review that have been left unresolved, and some of the new text could use a bit of cleaning up. My suggested edits:
Introduction
Line 60: Strangely, this actually makes less sense than the original wording. Metabarcoding is often conducted using ribosomal RNA data, so what is the “combination” referred to here? “More complete” than what—metabarcoding without 18S rDNA?
Line 66-67: “…have received increasing attention.”
Materials and Methods
Line 139: “RDP classifier was used…” I can only assume the “comparison threshold” here refers to the “confidence threshold” of RDP classifier, for which the default is 80%. Why was this threshold lowered for this study?
Results
Figure 3 caption “D is winter..”
Discussion
Line 319-322: This isn’t any stronger an argument for the effects of wind strength. Correlation does not equal causation.
Line 340: perhaps it might be simpler to say “…which reflects that the relationships between species is more complex.”
Lines 357-358: “In particular, the concentration of NH4+-N is telling, as this nitrogen species is the most directly absorbed by algae [insert citation].”
Line 358: “In a similar study, it was reported that Discostella cell density was high in the late stage of freezing, reaching a peak in August in the Jordan Pond, USA [42].” Were the nutrient swings (particularly nitrogen) the same in that study as well?
Line 364: “By analysis of the diatom sediment cores in Lake Tahoe, USA, it was found that…”
Lines 389-403: Much better—hard data makes for better comparisons.
Line 392: “…D. nipponica (96.60%) is 3-4 um in cell diameter [45].”
Line 394: I wouldn’t consider an increase of 500 nm at the low end of the range as enough to consider a jump to “large-celled”, considering some diatom species can be hundreds of um in diameter and length.
“…community was mainly composed of A. granulata…” Be wary of AutoCorrect
Line 395: “This study…” (Capitalize for new sentence)
Line 397: “…were found to be something between the…” I’m assuming the authors mean the “characteristics” (whatever they are…) suggest a community profile between the temperate and tropical zones?
Line 414: “…high-throughput sequencing techniques…”
Line 417: “…and diatoms (mainly Nitzschia) were dominant in the Danjiang River [48]. After the…” Spacing and capitalization. Only “Chlorophyta” would be capitalized, as that is the only taxonomic group—dinoflagellates, diatoms and cyanobacteria are all less formal terms. Check throughout the rest of the paragraph.
Lines 425-427: As I mentioned in the previous review:
“Additionally, several of the taxa identified in this study and listed in the Supplementary materials are taxa associated with marine environments (Pseudo-nitzschia, Skeletonema, Fragilariopsis, Licmophora, Actinocyclus)—were these actual “detected species” or the result of some sort of artifact of the eDNA and metabarcoding techniques? Is “species detection” really “improved” if the incorrect species are identified?”
Characterizing a phytoplankton assemblage is about more than raw species counts—context matters in identification
Lines 433-442: The authors have misconstrued my comments on Cyclotella versus Discostella. The fact that Discostella wasn’t identified in the reservoir prior to the description of the genus has nothing to do with misidentification. If the earlier studies counted, for example, Cyclotella woltereckii in 1992 and 1993, it doesn’t mean they misidentified Discostella woltereckii—Cyclotella woltereckii was the accepted name at the time. What Cyclotella species were identified in the earlier studies—were they species which were not later transferred to Discostella?
Author Response
Thank you for taking time out of your busy schedule to review the manuscript. I benefit from these comments in the process of correcting the manuscript. Now we have carefully corrected and replied the manuscript for this revision to meet with the published standard in journal water. All revised portions are indicated in the text by highlighting in the revised manuscript, and language of revised manuscript has been improved. The revision instructions are as follows:

Round 3
Reviewer 1 Report
This version is much better. There are still a few typos ("A. granulate" instead of "A. granulata") and missing spaces between words, but these can be worked out with the editorial staff of the journal.
As for Discostella vs. Cyclotella, I suppose technically it is correct that this is the first report of Discostella in this reservoir...if only because the genus did not exist at the time of the previous studies. I am still curious to know if the Discostella basionym taxa indicated in this study were present in the older studies, though....